# Sex-Dependent Differential Expression of Lipidic Mediators Associated with Inflammation Resolution in Patients with Pulmonary Tuberculosis

**DOI:** 10.3390/biom12040490

**Published:** 2022-03-24

**Authors:** Claudia Carranza, Laura Elena Carreto-Binaghi, Silvia Guzmán-Beltrán, Marcela Muñoz-Torrico, Martha Torres, Yolanda González, Esmeralda Juárez

**Affiliations:** 1Laboratorio de Inmunobiología de la Tuberculosis, Instituto Nacional de Enfermedades Respiratorias Ismael Cosío Villegas, Calzada de Tlalpan 4502, Sección XVI, Mexico City 14080, Mexico; claudia.carranza@iner.gob.mx (C.C.); lecarreto@iner.gob.mx (L.E.C.-B.); mtorres@iner.gob.mx (M.T.); 2Departamento de Investigación en Microbiología, Instituto Nacional de Enfermedades Respiratorias Ismael Cosío Villegas, Calzada de Tlalpan 4502, Sección XVI, Mexico City 14080, Mexico; sguzman@iner.gob.mx (S.G.-B.); ygonzalezh@iner.gob.mx (Y.G.); 3Servicio Clínico de Tuberculosis, Instituto Nacional de Enfermedades Respiratorias Ismael Cosío Villegas, Calzada de Tlalpan 4502, Sección XVI, Mexico City 14080, Mexico; dra_munoz@iner.gob.mx

**Keywords:** eicosanoids, inflammation, oxidative stress, pulmonary tuberculosis, sex-dependent

## Abstract

There is a sex bias in tuberculosis’s severity, prevalence, and pathogenesis, and the rates are higher in men. Immunological and physiological factors are fundamental contributors to the development of the disease, and sex-related factors could play an essential role in making women more resistant to severe forms of the disease. In this study, we evaluated sex-dependent differences in inflammatory markers. Serum samples were collected from 34 patients diagnosed with pulmonary TB (19 male and 15 female) and 27 healthy controls (18 male and 9 female). Cytokines IL2, IL4, IL6, IL8, IL10, IFNγ, TNFα, and GM-CSF, and eicosanoids PGE2, LTB4, RvD1, and Mar1 were measured using commercially available immunoassays. The MDA, a product of lipidic peroxidation, was measured by detecting thiobarbituric-acid-reactive substances (TBARS). Differential inflammation patterns between men and women were observed. Men had higher levels of IL6, IL8, and TNFα than women. PGE2 and LTB4 levels were higher in patients than healthy controls, but there were no differences for RvD1 and Mar1. Women had higher RvD1/PGE2 and RvD1/LTB4 ratios among patients. RvD1 plays a vital role in resolving the inflammatory process of TB in women. Men are the major contributors to the typical pro-inflammatory profile observed in the serum of tuberculosis patients.

## 1. Introduction

Tuberculosis (TB) is caused by *Mycobacterium tuberculosis* (Mtb), and mainly affects the lungs. Lung Mtb infection induces the activation of alveolar macrophages, which are the main drivers of the innate immune response in the lung. Macrophages engulf, process, and present the antigens to T lymphocytes, triggering an adaptative immune response [1]. Although only 5% to 10% of individuals exposed to Mtb develop active TB, 70% of those who develop it are men [2]. Moreover, incidence rates, lung damage, coinfections, smear positivity, and mortality rates are higher in men [3,4]. Of the estimated 9.9 million people infected with Mtb in 2020, men comprised 53% of the deaths among HIV-negative people, women 32%, and children 15% of the deaths. Among people living with HIV, men comprised 50% of the deaths [5].

Immunological and physiological factors may cause differences in males’ susceptibility to developing disease post-exposure to Mtb [2,6,7]. Differential immune responses to foreign and self-antigens in males and females include differences in the abundance of the Toll-like receptor 7 (TLR7) gene, which is encoded on the X chromosome, and the effects of the androgen response elements (AREs) and estrogen response elements (EREs) on genes of innate immunity, which cause males to produce more tumoral necrosis factor-alpha (TNFα) and interleukin-10 (IL10) than females [8]. In this study, we hypothesized that such disparity could be explained, at least in part, to differences between sexes in the inflammatory mediators of the eicosanoid’s pathway. Little is known about sex-related factors intervening in the eicosanoids’ pathways in TB. However, the level of eicosanoids in plasma varies between clinical states of infection, depicting an imbalance in eicosanoids in active TB patients. In addition, murine models revealed a critical role for PGE2 and other eicosanoids in susceptibility to Mtb infection, which correlated with reduced antigen-specific T-cell responses and increased levels of regulatory T cells; and allergy models suggest the eicosanoids’ involvement in the regulation of inflammatory responses in lungs, which promotes vasodilation, vascular permeability, and bronchodilation [9,10,11,12,13], highlighting the relevance of the pathway.

Eicosanoid lipids and cytokines mediate the immune response to Mtb. Both alveolar macrophages and Mtb modulate inflammatory processes mainly through the production of pro-inflammatory cytokines, such as interferon-gamma (IFNγ), TNFα, and interleukin-1 (IL1) [11]. Eicosanoids, generated enzymatically through the oxygenation of omega-6 polyunsaturated fatty acids and arachidonic acid by cyclooxygenases and lipoxygenases, play a fundamental role in regulating inflammation. Their dysregulation, indeed, leads to chronic inflammation and extensive tissue damage [14]. These lipids can promote or resolve inflammation through the production of leukotrienes (LT), prostaglandins (PG), lipoxins (LX), resolvins (Rv), and maresins (Mar) [15]. Moreover, sex is believed to be a critical variable for regulating various lipoxygenases, potentially regulating the eicosanoid pathways [16].

Prostaglandin E2 (PGE2) belongs to a family of eicosanoids related to lipidic hormones. The role of PGE2 in hosts’ defense against bacterial infections has been clearly described [17]. PGE2 promotes a switch to an anti-inflammatory phenotype, inhibiting macrophage and neutrophil activity. Furthermore, PGE2 induces a Th2 or Th17 response, limiting T cell activation and proliferation. 

Unlike PG, leukotrienes (LT) promote the activation of alveolar macrophages [1] through the production of TNFα, which is stimulated by the pro-inflammatory leukotriene B4 (LTB4) but reduced by lipoxin A4 (LXA4). Proper control of inflammation in TB depends on a balanced production of eicosanoids. In addition to the modulation of TNFα production, eicosanoids influence inflammatory processes via cell regulation, apoptosis, and adaptative immunity [17]. LTB4 induces the release of lysosomal enzymes, neutrophil defensins, and nitric oxide production, enhancing the innate immune response [18]. 

Other lipidic mediators, derived from eicosapentaenoic and docosahexaenoic acids (components of omega-3 fatty acids) by the enzymatic action of lipoxygenases and cyclooxygenase-2, are involved in the resolution of inflammation [19]. Two of these molecules deserving further investigation are resolvin D1 (RvD1) and maresin 1 (Mar1), which promote the resolution of inflammation by reducing the production of PGE2 and LTB4 and activating macrophages’ bactericidal activity against Mtb [20]. In addition, mouse models of tuberculosis reported the role of LTB4 and PGE2 in bacterial clearance and the role of LXA4 and 15-epi-lipoxin A4 (15-epi-LXA4) in the hosts’ inflammatory response [21]; in tuberculosis patients, by contrast, increased levels of LXA4, 15-epi-LXA4, and PGE2 and decreased levels of LTB4 compared to healthy subjects have been found in plasma [22]. However, these processes’ relation to sex has not been investigated.

Better insight into how sex contributes to these disparities may help to unravel our understanding of the immune response in tuberculosis. In this study, we measured the plasmatic concentrations of several inflammatory cytokines and lipidic mediators in both patients with active TB and healthy subjects, comparing the response between sexes to characterize the systemic inflammatory profile of the disease, and assessing whether the sex of the subjects defined characteristic patterns. 

## 2. Materials and Methods

### 2.1. Study Group

We conducted a prospective cross-sectional evaluation to determine immune-related inflammation biomarkers in patients with TB and their association with sex. The participants were recruited at the National Institute of Respiratory Diseases (INER) in Mexico City. The Institutional Ethics and Research Committee approved this study, and all participants provided written informed consent. We included 34 patients, 19 male and 15 female, recently diagnosed with pulmonary TB or multidrug-resistant TB who started a new therapeutic regime. All patients were diagnosed with active TB after a positive sputum smear test or Xpert^®^ MTB/RIF assay that was further confirmed by *M. tuberculosis* culture; patients were recruited within the first ten days of anti-TB treatment. We included 27 healthy controls, 18 male and 9 female. All study participants were between 18 and 65 years of age. Individuals with HIV infection, asthma, chronic inflammatory diseases, or cancer were excluded.

Disease severity was evaluated in terms of the smear report, laterality of the lung lesions, presence of cavities, and the score of radiographical abnormalities (SRA), as reported elsewhere [23,24]. The SRA determined the presence, distribution, and extent of consolidation, fibrosis, lung distortion, bronchiectasis, and parenchymal abnormalities in four quadrants; each quadrant was scored from 0 to 5, where 0 indicated a normal appearance, and 5 indicated severe abnormality. The maximum score was 20 and represented a higher lung parenchyma involvement. The number of quadrants with lesions determined the lung damage extent. The healthy controls had typical chest X-ray images and laboratory results. 

### 2.2. Serum Collection and Processing

Blood samples were obtained from the participants, and the serum was retrieved via centrifugation, aliquoted, and stored at −70 °C until use. The aliquots intended to quantitate lipids (eicosanoids and estradiol) underwent extraction using ethanol precipitation before analysis, and were stored at −20 °C until use. 

### 2.3. Immunoassays

Cytokines were measured using an 8-plex bioplex system (Bio-Rad, Hercules, CA, USA) for IL2, IL4, IL6, IL8, IL10, IFNγ, TNFα, and GM-CSF according to the manufacturers’ instructions. The concentrations of the eicosanoids PGE2, LTB4, RvD1, and Mar1 and estradiol were quantified using commercial EIA kits (Cayman Chemical, Ann Arbor, MI, USA). All assays were performed according to the manufacturer’s instructions. Samples were analyzed in duplicate, and optical density was determined at 450 nm using a MultisKan Ascent microplate reader (Thermo Fisher Scientific, MA, USA). MDA, a lipid peroxidation product, was measured with the thiobarbituric-acid-reactive substance (TBARS) method using a TBARS assay kit (Cayman Chemical) according to the manufacturer’s instructions.

### 2.4. Statistics

Categorical data are presented as individual results or box plots with median and quartiles; frequencies, percentages, and continuous data are presented as median and range. The categorical variables between study groups were compared using Fisher’s exact test; the continuous data were compared using non-parametrical statistical methods, the Kruskal–Wallis test with Dunn’s post-hoc for multiple comparisons, the Mann–Whitney U test for comparison of unpaired data, and the Wilcoxon rank test for matched pairs. Spearman’s Rho correlation was used to establish the correlation between variables. Analyses were performed using GraphPad Prism version 9.0 (GraphPad Software, La Jolla, CA, USA). 

## 3. Results

### 3.1. Clinical Characteristics of the Participants

We included 34 patients with pulmonary TB and 27 healthy subjects. The patients showed demographic and laboratory data typical of their disease, and the healthy control had normal parameters (Appendix A). Demographic and clinical variables disaggregated by sex are listed in Table 1. When comparing women and men, we found no differences in age, BMI, AINES consumption, smoking history, diabetes, hypertension, or white blood cell counts. The healthy controls exhibited the characteristic sex-dependent differences in red blood cell counts, but they were lost in the patients. Serum creatinine and uric acid normal values are typically higher in men [25,26]. As expected, creatinine was significantly higher but within the normal range in both the male patients and male healthy controls. Uric acid, a natural antioxidant, was higher in males, yet within the normal range, only in the healthy controls group; however, this sex-dependent feature was lost in the patients. Men had more radiographic abnormalities and were more likely to present cavities; however, this did not reach significance (Table 2). 

Among the laboratory parameters, monocyte to lymphocyte (M/L) ratio, neutrophil to lymphocyte (N/R) ratio, and red blood cell distribution width % (RDW%) have been described as systemic inflammatory markers in patients with tuberculosis [27,28]. We analyzed these parameters to determine their relevance in sex. All three markers were significantly elevated in patients (Figure 1a); when disaggregating the data by sex, the M/L ratio was higher in men, but only the N/L ratio was significantly higher in men than women (Figure 1b).

### 3.2. Cytokines

We observed the typical high amounts of circulating IL6, IL8, and TNFα in the patients (Figure 2a–c, left). When disaggregating the results by sex, we observed that men in this group provided the more significant contribution to this well-known profile. Men had significantly higher IL8 and TNFα levels in the group of patients with TB (Figure 2a–c, center), but not in the healthy controls (Figure 2a–c, right). The circulating IL6 was also higher in the male patients, but it did not reach significance. We found no differences in the levels of IL10 (Figure 2d) or the other cytokines (Appendix A) in any group. Surprisingly, the levels of IL8 and TNFα of the female patients were similar to those of the healthy female controls, and only the levels of IL6 were higher in the female patients (Appendix A).

### 3.3. Eicosanoids

Furthermore, we analyzed the serum PGE2, LTB4, RvD1, and Mar1 levels. We found higher pro-inflammatory PGE2 and LTB4 levels in the patients (Figure 3a,b, left). In contrast, serum concentrations of RvD1 and Mar1 were similar in patients and healthy controls (Figure 3c,d, left). The pro-inflammatory PGE2 level was significantly higher in the male patients but not in the healthy controls (Figure 3a, center). We found no differences in the levels of the rest of the eicosanoids between women and men in either group (Figure 3a–d, center, right).

Because disease severity in TB has been reported to be associated with an increased ratio of LXA4/LTB4 and a reduced ratio of PGE2/LXA4, rather than changes in absolute levels of specific metabolites [22], and the relative contribution of the eicosanoids is relevant to the complete resolution of inflammation [29], we investigated the following pro-resolving/pro-inflammatory eicosanoid ratios: Mar1/PGE2, Mar1/LTB4, RvD1/PGE2, and RvD1/LTB4, as well as Mar1/RvD1 and PGE2/LTB4 ratios. We found no differences in the Mar1/PGE2 ratio in any case (Figure 4a); however, the Mar1/LTB4 ratio was significantly higher in the healthy subjects, suggesting their ability to resolve inflammation (Figure 4b, left). The RvD1/PGE2 and RvD1/LTB4 ratios were significantly higher in the female patients, suggesting women’s ability to resolve inflammation (Figure 4c,d, center). Despite having higher pro-resolutory lipids relative contributions, female patients had significantly lower RvD1 levels relative to Mar1 (Figure 5a), suggesting a less relevant role for Mar1. We found no differences in the PGE2/LTB4 ratio in any case (Figure 5b). Interestingly, there were no sex-dependent differences in the eicosanoid profiles in the healthy control group.

### 3.4. Systemic Marker of Lipid Peroxidation

Because we found no explicit lipid profiles preferentially associated with sex, we investigated whether the lipid peroxidation was a factor in the inability of pro-resolutory lipids to resolve inflammation. MDA is a dialdehyde, a highly reactive hydroperoxide, and a good indicator of lipoperoxidation. The serum from patients with TB had significantly higher levels of MDA than the healthy controls, but its levels were not dependent on sex in either group (Figure 6a) despite having a moderate positive correlation with the other pro-inflammatory markers PGE2 and LTB4 (Figure 6b).

### 3.5. Correlation of Pro-Inflammatory Markers with the Severity of the Disease

We explored the immunological implications of sex-dependent differences in inflammatory markers on the severity of the disease. Our cohort was not large enough to conclusively detect the effects of sex on the overall inflammation markers under investigation. However, a principal components analysis (PCA) that included all inflammatory parameters, estradiol, and sex (but not clinical parameters) demonstrated clustering of females in PC2, and males were distributed along PC1 (Figure 7a). PC1 (38.96%) was mainly influenced by IL8, PGE2, RvD1, Mar1, and LTB4; PC2 (21.34%) was mainly influenced by IL6 and TNFα (Figure 7b). Afterward, we performed a correlation matrix of the inflammatory parameters of the PC1 with the clinical parameters age, BMI, smear grade, and lung damage extent. We observed differential correlation patterns between women and men (Figure 7c,d). Most of the positive correlations among the inflammatory biomarkers were expected for men because of the selection based on PC1. Age exhibited no significant correlation with the inflammation parameters for either women or men (*p* > 0.05). Interestingly, women showed no correlation between any inflammatory metabolites and the smear grade, but MDA and PGE2 had a significant positive correlation with the extent of lung damage. In men, however, age positively correlated with BMI, and PGE2 and RvD1 negatively correlated with lung damage extent (*p* < 0.05). PGE2 and RvD1 correlation plots are depicted in Figure 7e,f.

## 4. Discussion

The increased susceptibility to the development of pulmonary tuberculosis in men has been associated with social and epidemiological factors such as reporting bias and cultural, social, and economic factors (i.e., men have higher rates of smoking and HIV, more social contacts associated with work, and higher rates of health care service underutilization) [6,30,31]. Other social and epidemiological factors, such as low income, low education, and high alcohol consumption, are associated with treatment failure in TB and multi-drug resistant TB (TB-MDR), but they are not attributed to men [32]. On the other hand, social and epidemiological factors influence exposure to TB, and immunological and physiological factors may cause differences between men’s and women’s susceptibility to developing disease post-exposure [2,6,33]. Sex-specific immune responses that likely contribute to the pathogenesis of pulmonary tuberculosis include poorer specific T cell-dependent IFNγ production and lower antibody responses in men [7]. However, the lipid-dependent and other inflammatory pathways intervening in the development of severe forms of the disease in men have not been explored.

When analyzing the clinical parameters disaggregated by sex, we observed laboratory findings typical of tuberculosis patients (i.e., red blood cells < 4.3 × 10^6^/mm^3^, hematocrit < 40%, leukocytosis, neutrophilia, and lymphopenia) [34,35]. Interestingly, the clinical laboratory parameters that typically differ between men and women (such as erythrocyte counts, hemoglobin, hematocrit, and uric acid) were not different between male and female TB patients, mainly because most of these laboratory features, which are usually higher in men, were diminished in the male patients. This finding suggests extended damage to other organs beyond the pulmonary TB infection, such as bone marrow and kidney, which is probably related to Mtb dissemination [36]. 

The clinical biomarkers of systemic inflammation, the M/L ratio, N/L ratio, and RDW (%) previously reported for TB patients, may predict poor outcomes in other diseases [37,38,39]. In our study, we found that these markers increased in patients with pulmonary TB. We found that the N/L and M/L ratios in TB patients were higher in men than those in women, indicating that men had a higher inflammatory condition that could develop into more severe clinical forms. Higher N/L ratios (>2.5) are associated with susceptibility to developing active TB, a need for retreatment, delay in smear-negative conversion, pulmonary cavitation, and mortality [40,41,42]. Higher M/L ratios (>0.5) are associated with susceptibility to develop active TB, tuberculosis pleuritis, inability to control infection, and higher pathology scores [43,44,45]. A reduction in these ratios is expected to occur in patients who respond successfully to treatment. Routine assessment of these hematological parameters may provide valuable data to complement conventional measures for assessing disease conditions. The lack of differences between women and men in other laboratory and clinical features, such as BMI, diabetes, smoking, and hypertension, suggests that they do not account for the sex-dependent pro-inflammatory profile. Thus, we evaluated inflammatory immunological parameters.

Pro-inflammatory cytokines are typically increased in patients with tuberculosis [46,47]. When investigating whether a set of inflammatory and pro-resolutory effectors are differentially expressed in men and women with TB, we found sex-dependent differences in the cytokine profiles in the group of patients. Men had higher IL6, IL8, and TNFα levels than women, and female patients’ and healthy controls’ IL8 and TNFα levels were similar, supporting the notion that men are the main contributors to the highly pro-inflammatory profile observed in pulmonary TB patients. Despite the marked differences in the pro-inflammatory cytokine patterns, we found no differential IL10 or IL4 expression.

Regarding eicosanoids, we observed elevated levels of PGE2 in men compared with women, and no differences in the pro-resolutory RvD1 and Mar1, suggesting the patients’ inability to resolve inflammation. Mouse models indicated that during the early phase of the infection low concentrations of PGE2 are compatible with temporal bacterial control and the regulation of LTB4 and inflammation. However, during the late stages of the disease, PGE2 rises and exerts immunosuppressive functions that reduce the cellular immune responses and promote pneumonia [48,49]. Our patients had a long history with their disease; thus, the higher concentrations of PGE2 are likely pro-inflammatory, and the male population largely contributed to the pro-inflammatory profile of the group (Figure 8a). Despite making a higher pro-resolutory lipids relative contribution, female patients had a significantly lower Mar1/RvD1 ratio than men, suggesting a more prominent role for RvD1 in resolving inflammation in TB.

The male patients had marginally lower pro-resolutory lipids levels than females, suggesting a defect in function rather than quantities. We investigated the lipid peroxidation levels as an indirect way to measure the inability of these lipids to perform their regulatory actions. The lipid peroxidation levels were higher in patients than those in healthy controls, but were not associated with sex. However, the moderate correlations with LTB4 and PGE2 that MDA exhibited were mainly derived from the male population. Thus, men may have more lipid peroxidation rates that also prevent resolution.

Although men have been reported to have a more considerable extent of pulmonary disease [50], we did not find significant differences between men and women in the severity of the disease parameters, probably due to the different disease stages at the sampling time. However, the principal component analysis and the correlation matrix suggest a sex bias in the disease’s form. The negative correlation between PGE2 and RvD1 and the extent of lung damage in males suggest an inability of men to resolve inflammation, which may contribute to lung damage (Figure 8b).

This study did not find sex-dependent differences in the cytokines and eicosanoid profiles of the healthy control group. The lack of sex-dependent metabolites expression in the healthy controls suggests that sex-related factors (i.e., hormones and miRNAs) are not the sole contributors to the male patients’ differential pro-inflammatory characteristics. Other sex-related factors must be associated with the infection in humans, and are worthy of further study. The pro-inflammatory sex bias in pulmonary TB has implications for therapy, interventions designed to preserve lung function, and clinical follow-up programs. Furthermore, it appears that drug efficacy, metabolism, and immune response may also be sex-influenced, causing variabilities in host-directed therapy for TB [51]; these outcomes would make sex disparities even wider.

The small number of enrolled subjects limited this study; it is necessary to test the hypothesis in larger cohorts. Our results highlight the need for longitudinal studies to answer questions regarding the outcome: does the excess of inflammation in men result in more lung damage and respiratory dysfunction after treatment? Do men need more time to achieve a negative smear test? Does the differential inflammatory profile directly impact healing and mortality rates? What are the implications for women at postmenopausal ages?

## 5. Conclusions

Men were the major contributors to the typical pro-inflammatory profile observed in the serum of tuberculosis patients. The sex-dependent pro-inflammatory features that were significantly overrepresented in the male patients may have a role in disease severity, and they may potentially modify the knowledge generated from therapy and biomarkers research. A better understanding of how sex-dependent inflammatory response influences clinical outcomes could lead to a differentiated approach to treatment and follow-up, including the choice and duration of therapeutic regimen, the definition of treatment failure, and the interpretation of biomarkers associated with the response.

## Figures and Tables

**Figure 1 biomolecules-12-00490-f001:**
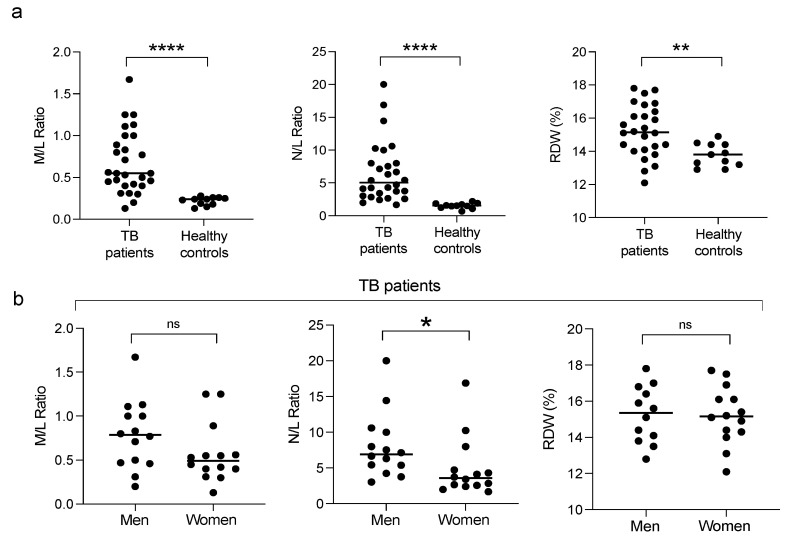
Hematological parameters of inflammation in pulmonary tuberculosis patients. Monocyte to lymphocyte (M/L) ratio, neutrophil to lymphocyte (N/L) ratio, and red blood cell distribution width % (RDW%) were calculated in patients and healthy controls (**a**). Data were disaggregated by sex for the patients (**b**). Individual results with medians are depicted. * *p* < 0.05, ** *p* < 0.01, **** *p* < 0.0001, ns = not significant.

**Figure 2 biomolecules-12-00490-f002:**
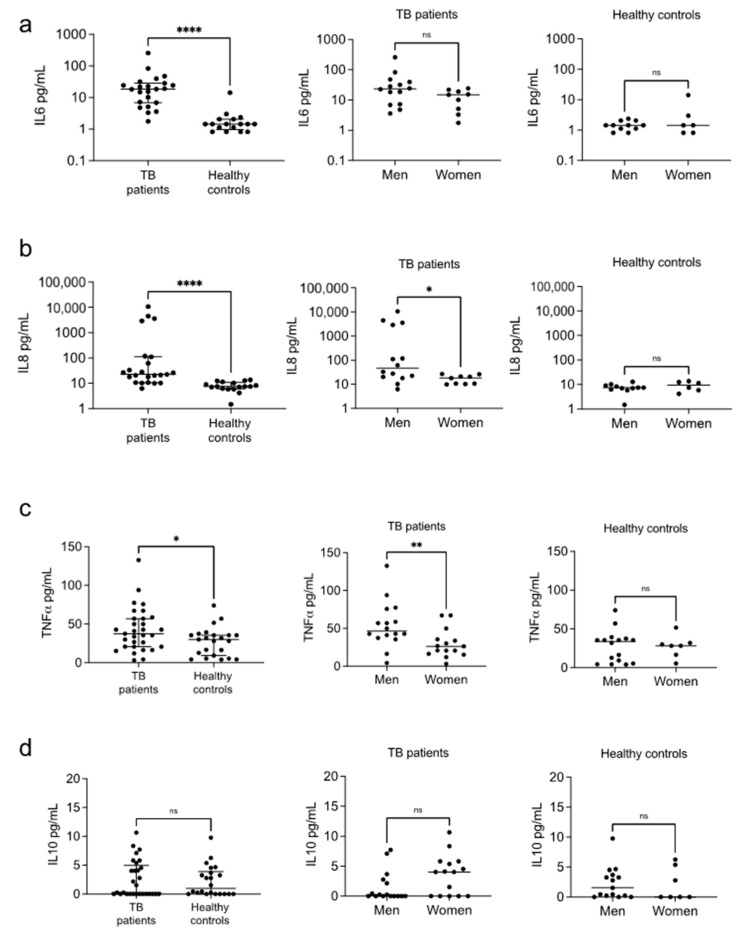
Circulating levels of cytokines. Serum levels of IL6 (**a**), IL8 (**b**), TNFα (**c**), and IL10 (**d**) were measured in patients with pulmonary tuberculosis and healthy controls using a Bioplex system. Individual results with medians and interquartile ranges are depicted (**left**). Data were disaggregated by sex for both the patients (**center**) and controls (**right**); individual results with medians are depicted. * *p* < 0.05, ** *p*<0.01, **** *p* < 0.0001, ns = not significant.

**Figure 3 biomolecules-12-00490-f003:**
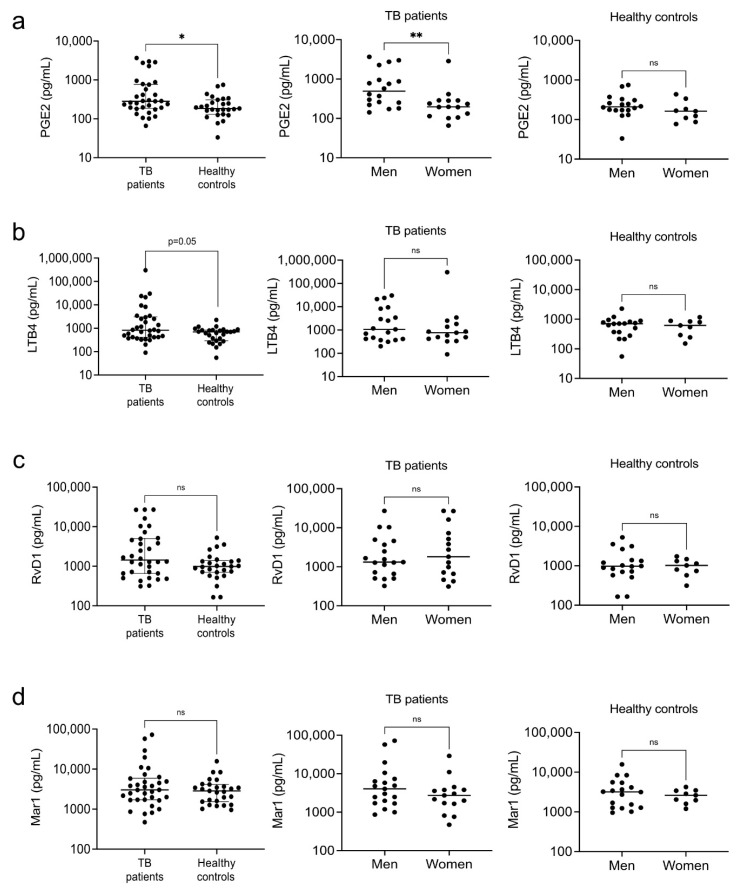
Circulating levels of eicosanoids. Serum levels of proinflammatory PGE2 (**a**) and LTB4 (**b**) eicosanoids and pro-resolutory RvD1 (**c**) and Mar1 (**d**) eicosanoids were measured in patients with pulmonary tuberculosis and healthy controls by ELISA. Individual results with medians and interquartile ranges are depicted (**left**). Data were disaggregated by sex for both the patients (**center**) and controls (**right**); individual results with medians are depicted. * *p* < 0.05, ** *p* < 0.01, ns = not significant.

**Figure 4 biomolecules-12-00490-f004:**
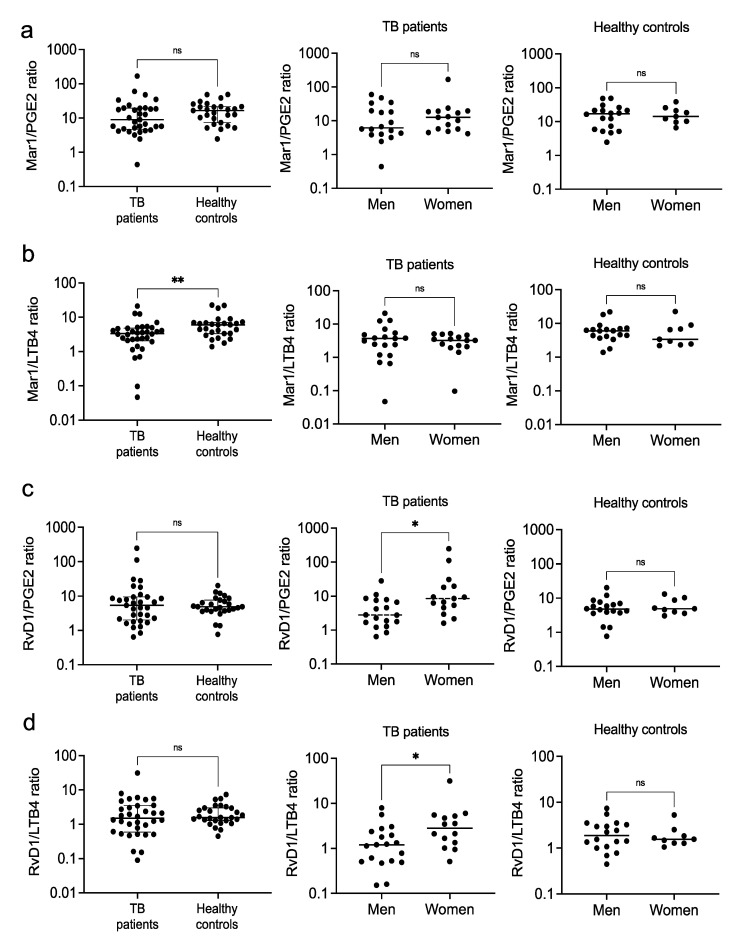
Pro-resolutory/pro-inflammatory ratios of circulating eicosanoids. Mar1/PGE2 (**a**), Mar1/LTB4 (**b**), RvD1/PGE2 (**c**), and RvD1/LTB4 (**d**) ratios were calculated from the serum levels of eicosanoids of patients with pulmonary tuberculosis and healthy controls. Individual results with medians and interquartile ranges are depicted (**left**). Data were disaggregated by sex for both the patients (**center**) and controls (**right**); individual results with medians are depicted. * *p* < 0.05, ** *p* < 0.01, ns = not significant.

**Figure 5 biomolecules-12-00490-f005:**
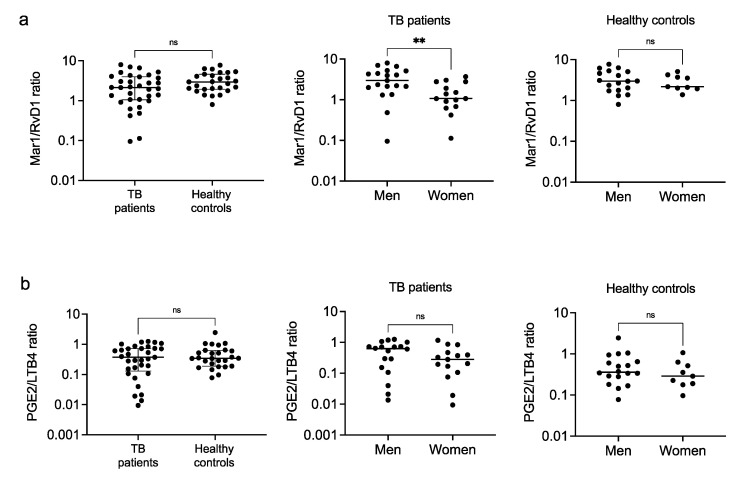
Pro-resolutory and pro-inflammatory ratios of circulating eicosanoids. Pro-resolutory Mar1/RvD1 (**a**), and pro-inflammatory PGE2/LTB4 (**b**) ratios were calculated from the serum levels of eicosanoids of patients with pulmonary tuberculosis and healthy controls. Individual results with medians and interquartile ranges are depicted (**left**). Data were disaggregated by sex for both the patients (**center**) and controls (**right**); individual results with medians are depicted. ** *p* < 0.01, ns = not significant.

**Figure 6 biomolecules-12-00490-f006:**
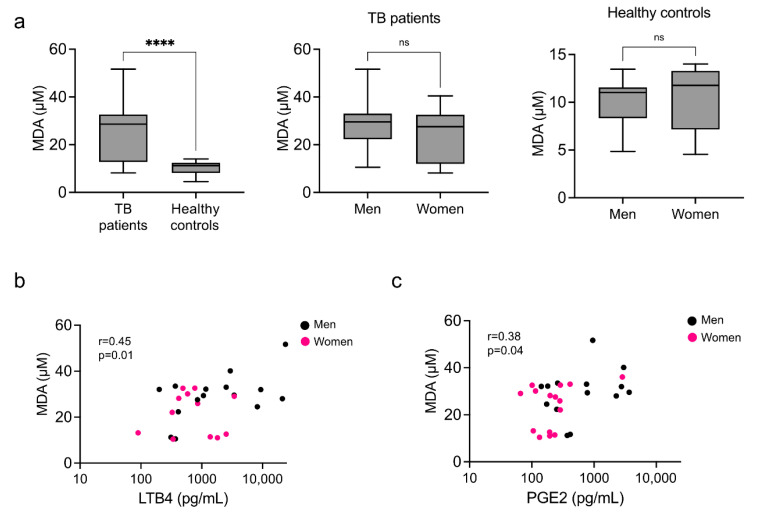
Circulating levels of malonaldehyde (MDA). Serum levels of MDA were measured in patients with pulmonary tuberculosis and healthy controls by ELISA (**a**). Box plots with medians and quartiles are depicted (**left**). Data were disaggregated by sex for both the patients (**center**) and controls (**right**). **** *p* < 0.0001, ns = not significant. Correlation plots for MDA and pro-inflammatory eicosanoids, LTB4 (**b**) and PGE2 (**c**) are depicted. Spearman’s rho with *p* values are included within the plots.

**Figure 7 biomolecules-12-00490-f007:**
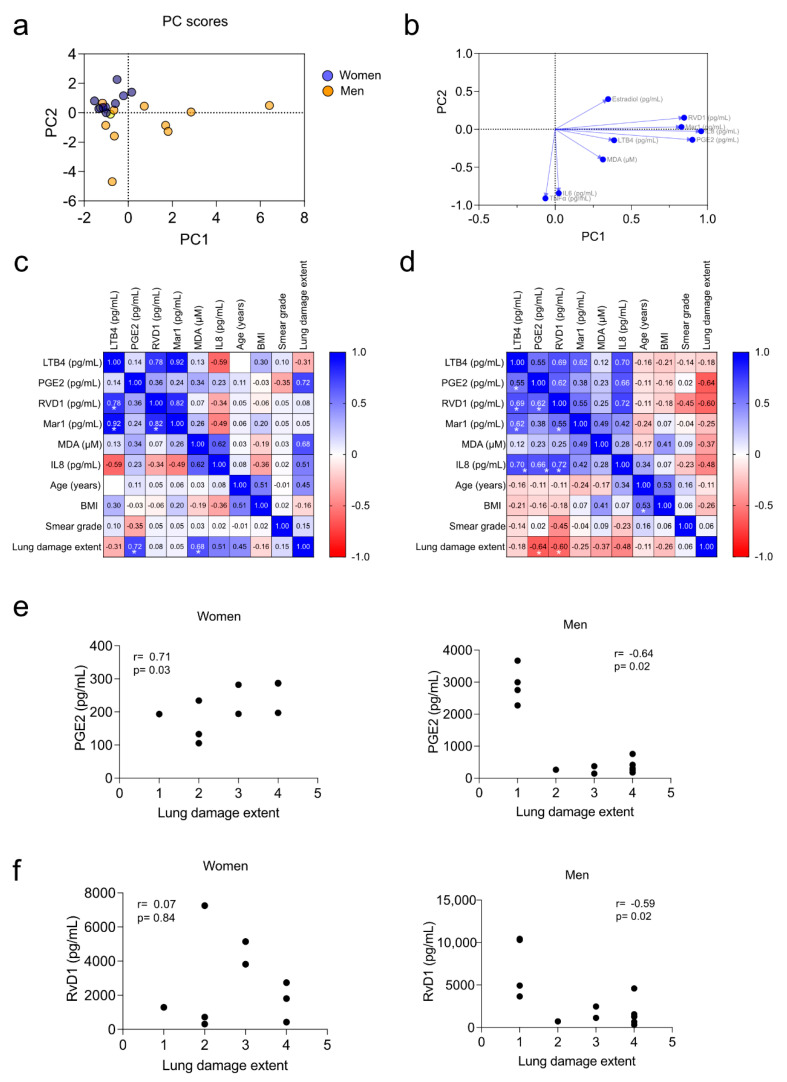
PGE2 and RvD1 negatively correlated with lung damage in men. Principal components (PC) analysis was performed using the following variables: sex, PGE2, LTB4, RVD1, Mar1, estradiol, MDA, TNFα, IL8, IL6, RvD1/PGE2 ratio, RvD1/LTB4 ratio, and Mar1/RvD1 ratio. Patients with missing data were excluded. PC Scores (**a**) and PC Loadings (**b**) are depicted. Spearman’s correlation matrix was performed for the parameters influencing the PC1 and the clinical variables age, BMI, smear grade, and lung damage extent for both women (**c**) and men (**d**). Rho’s values are depicted within the heat maps; * indicates unadjusted *p* < 0.05 annotated only in the inferior half of the matrix. Correlation plots of PGE2 (**e**) and RvD1 (**f**) with lung damage extent in women and men are depicted.

**Figure 8 biomolecules-12-00490-f008:**
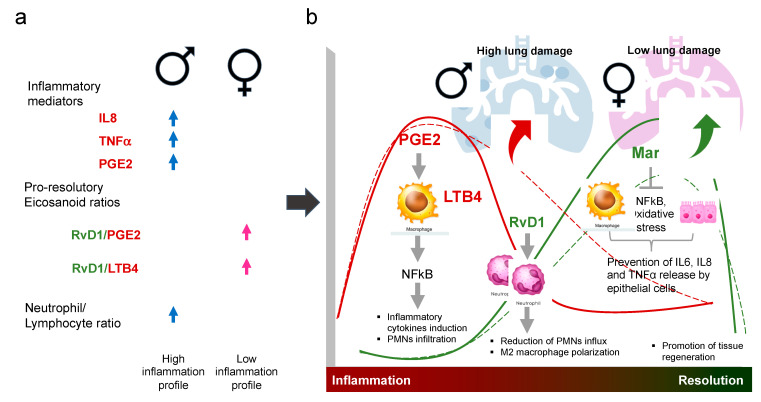
Sex-dependent alterations in inflammation/resolution pathways in active tuberculosis patients. (**a**) Men exhibited a circulating high inflammation profile, whereas women had a pro/resolution profile. (**b**) Implications of our findings. Pro-inflammatory eicosanoids PGE2 and LTB4 induce the production of pro-inflammatory cytokines, which results in neutrophils infiltration. The subsequent production of pro-resolution eicosanoids RvD1 and Mar1 helps to control the cytokine production, reduce the neutrophil infiltration and oxidative stress, and promote tissue regeneration. Dotted lines indicate the more pro-inflammatory profile observed in men, which makes them more susceptible to develop lung damage. The more pro-resolution profile in women (solid lines) may be protective of the lungs.

**Table 1 biomolecules-12-00490-t001:** Demographic and clinical characteristics of the participants.

	Healthy Controls			Patients		
Characteristic	Women, *n* = 9	Men, *n* = 18	*p* Value	Women, *n* = 15	Men, *n* = 19	*p* Value
Age, years, median (range)	33 (22–62)	31 (23–58)	0.7328	43 (19–62)	47 (18–64)	0.8304
BMI, median (range)	23.8 (20–27.5)	24.1 (20.7–30)	0.8503	21.8 (12.7–39.6)	21.7 (15.8–33.3)	0.7320
AINES consumption the previous week, *n* (%)	1 (11.1)	0 (0)	0.3333	6 (40)	7 (36.8)	>0.9999
Smoking history, *n* (%)	3 (33.3)	2 (40)	0.2950	0 (0)	2 (10.5)	0.4920
Diabetes, *n* (%)	0 (0)	0 (0)	>0.9999	9 (60)	11 (57.8)	>0.9999
Hypertension, *n* (%)	0 (0)	0 (0)	>0.9999	1 (6.7)	1 (5.2)	>0.9999
Leukocytes (10^3^/mm^3^), median (range)	8 (3.6–10.6)	6.3 (4.5–7.7)	0.0681	8.4 (4.1–15)	9.6 (5.9–15.2)	0.1259
Neutrophils (10^3^/mm^3^), median (range)	4.3 (1.2–7.6)	3.5 (2.1–4.7)	0.0816	5.7 (2.2–13.5)	7.3 (4.4–13)	0.0569
Lymphocytes (10^3^/mm^3^), median (range)	2.4 (1.6–3)	2.2 (1.3–5.3)	0.8838	1.7 (0.8–2.4)	1.2 (0.5–2.4)	0.0586
Monocytes (10^3^/mm^3^), median (range)	0.5 (0.4–0.5)	0.5 (0.3–0.7)	0.8848	0.8 (0.5–1)	0.95 (0.1–1.5)	0.3173
Eosinophils (10^3^/mm^3^), median (range)	0.2 (0–0.4)	0.1 (0.1–0.5)	0.9000	0.1 (0–0.2)	0.05 (0–10.1)	0.4900
Basophils (10^3^/mm^3^), median (range)	0 (0–0.1)	0 (0–0.1)	>0.9999	0 (0–0.1)	0.05 (0–0.2)	0.4484
Erythrocytes (10^6^/mm^3^), median (range)	4.6 (4.2–4.8)	5.4 (5–6)	<0.0001	4.2 (3.8–5.1)	4.4 (2.9–5.5)	0.8291
Hemoglobin (gr/dL), median (range)	13.7 (12.6–15.8)	16.1 (15.1–17.5)	<0.0001	12.4 (9.4–16.7)	12.6 (7.9–17.7)	0.9795
Hematocrit (%), median (range)	41.1 (37.7–45.6)	48.8 (42.9–52.1)	<0.0001	37.7 (27.9–48.8)	37.7 (24–52.8)	0.8574
Platelets (10^3^/mm^3^), median (range)	234 (157–379)	219 (151–320)	0.7512	344 (133–525)	327 (152–662)	0.9387
Glucose (mg/dL), median (range)	93.3 (86.7–109.9)	96 (81–117)	0.8427	128 (84–274)	138 (41–525)	>0.9999
Urea (mg/dL), median (range)	21.3 (12.84–36.4)	25.6 (17.12–42)	0.4990	23 (19–56.6)	27.8 (10.7–57)	0.5468
BUN (mg/dL), median (range)	9.9 (6–17)	12 (8–19.63)	0.5329	10.8 (9–26.4)	12.1 (5–23.9)	0.8377
Uric acid (mg/dL), median (range)	3.3 (3.3–3.3)	6.2 (4.5–10.4)	<0.0001	4.5 (2.6–7.9)	5.1 (1.4–13.69)	0.6848
Creatinine (mg/dL), median (range)	0.7 (0.5–0.8)	0.9 (0.7–1.2)	0.0006	0.5 (0.2–1.1)	0.8 (0.4–1.4)	0.0266

**Table 2 biomolecules-12-00490-t002:** Severity of the disease.

Characteristic	Women, *n* = 15	Men, *n* = 19	*p* Value
Bilateral, yes *n* (%)	7 (77.7)	8 (57.1)	0.3998
Cavities, yes *n* (%)	3 (33.3)	8 (57.1)	0.4003
Score of radiographic abnormalities, median (range)	8 (2–16)	10.5 (2–18)	0.6305
Smear grade, *n* (%)			
0	4 (26.7)	3 (15.7)	0.6722
1+	4 (26.7)	8 (42.1)	0.4764
2+	2 (13.3)	2 (10.5)	>0.9999
3+	5 (33.3)	3 (15.7)	0.4172
4+	0 (0)	3 (15.7)	0.2380
MDR, *n* (%)	6 (40)	7 (36.8)	>0.9999
Outcome, *n* (%)			
Unknown	7 (46.6)	8 (42.1)	>0.9999
Treatment success, *n* (% of known)	7 (87.5)	9 (81.8)	>0.9999

## Data Availability

The data used to support the findings of this study are included within the article.

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
