# Peer review of "Sex-Dependent Differential Expression of Lipidic Mediators Associated with Inflammation Resolution in Patients with Pulmonary Tuberculosis"

_biomolecules, 2022, doi:10.3390/biom12040490_

Round 1

Reviewer 1 Report

Carrozza and colleagues wrote an interesting paper. The idea research and the quality of presentation is high.

Below my minor suggestions:

  1. Introduction: updata data on TB wordwilde in according with TB Report 2021 (see and cite TB report 2021)
  2. Methods and results: well presented , tables and figures are clear
  3. Discussion: line 275 add the role of social determinations of health also in onset MDR and in worste outcome (see and cite Social determinants of therapy failure and multi drug resistance among people with tuberculosis: A review. Tuberculosis (Edinb). 2017 Mar;103:44-51). Furthermore discuss better the role of M/L ratio and pro-infiammatori citokine
  4. conclusion: give some proposal that came from your excellent paper

Reviewer 2 Report

The authors reported differences in inflammatory cytokines and lipid mediators between male and female patients with tuberculosis. I think the authors' point of view is very unique. I don't have any disagreement with their main argument. However, I would like to point out a few minor points.

  1. In “2.1 study group”, it is stated that chronic inflammatory diseases were excluded from the study group. I think you need to be specific about what diseases were excluded. Are patients with bronchial asthma, atopic dermatitis, periodontal disease, etc., as chronic inflammatory diseases that can occur at a young age, completely excluded from this study?

  1. In Table 1, only the gender differences among the healthy controls and patients are examined for significance. For the purpose of this study, I understood that only the gender difference was tested. However, in order to recognize the difference between the patient group and the healthy group, isn't it necessary to show the results of the significant difference test for both groups?

  1. There seems to be some template text left in lines 149-152. Please delete them.

  1. Figure 7a looks to me as unfinished. Please make it complete.

  1. Regarding the dysfunction of pro-resolutiory lipids in males described in the Discussion, is there any other literature that supports this hypothesis? I think the authors' hypothesis is important, but it seems a bit sudden to me.
